# A Truck-Borne System Based on Cold Atom Gravimeter for Measuring the Absolute Gravity in the Field

**DOI:** 10.3390/s22166172

**Published:** 2022-08-18

**Authors:** Helin Wang, Kainan Wang, Yunpeng Xu, Yituo Tang, Bin Wu, Bing Cheng, Leyuan Wu, Yin Zhou, Kanxing Weng, Dong Zhu, Peijun Chen, Kaijun Zhang, Qiang Lin

**Affiliations:** Zhejiang Provincial Key Laboratory of Quantum Precision Measurement, College of Science, Zhejiang University of Technology, Hangzhou 310023, China

**Keywords:** cold atom interference, cold atom gravimeter, truck-borne gravity measurement system, reservoir gravity measurement

## Abstract

The cold atom gravimeter (CAG) has proven to be a powerful quantum sensor for the high-precision measurement of gravity field, which can work stably for a long time in the laboratory. However, most CAGs cannot operate in the field due to their complex structure, large volume and poor environmental adaptability. In this paper, a home-made, miniaturized CAG is developed and a truck-borne system based on it is integrated to measure the absolute gravity in the field. The measurement performance of this system is evaluated by applying it to measurements of the gravity field around the Xianlin reservoir in Hangzhou City of China. The internal and external coincidence accuracies of this measurement system were demonstrated to be 35.4 μGal and 76.7 μGal, respectively. Furthermore, the theoretical values of the measured eight points are calculated by using a forward modeling of a local high-resolution digital elevation model, and the calculated values are found to be in good agreement with the measured values. The results of this paper show that this home-made, truck-borne CAG system is reliable, and it is expected to improve the efficiency of gravity surveying in the field.

## 1. Introduction

Atomic interference technology has been used to accurately measure some physical parameters such as the hyperfine structure constant [1,2], the universal gravitation constant [3], the rotation angular velocity [4,5], the gradient of the gravity field [6,7], the gravity acceleration [8,9,10] and the gravitational waves [11,12]. Moreover, atom interferometers are versatile tools for studying fundamental physics, such as testing the general relativity [13,14] and the equivalence principle [15,16], and measuring the weak force [17]. CAGs based on this technology have developed rapidly [18,19,20] since the first CAG prototype was realized in 1991 [21,22,23]. The CAG performances, including sensitivity, accuracy and long-term stability, have been improved in laboratory environments [10,19,24,25].

However, CAGs are not yet mature enough to maintain the same performance in the field. This research into the atomic gravimeter is still at the initial stage, and it will take a long time to transform the instrument from the laboratory prototype to the field engineering prototype. They are required to be more miniaturized, integrated and more stable, have a much lower power consumption, and be movable [26,27]. In recent years, researchers have carried out some field/outdoor experiments based on CAGs in stopped trucks [28,29], slow trucks [24], 0 g aircrafts [30,31], supersonic rockets [32] and space station [33]. For example, CAGs have been used to measure the absolute gravity values of different floors in one elevator and evaluate the vertical gradient of the gravity field [34]. They were also applied to absolute gravity survey [35] and geophysical observatories, and the long-term measurement stability of the CAG produced by Muquans company has reached below 1 μGal [36]. The significance of these field/outdoor research works indicates the following: (1) it can effectively evaluate the performance and indicators of the developed CAGs; (2) based on the scientific and technical problems encountered during the measurement process, some new engineering methods may be proposed; (3) it can extend CAGs to the civil field and serve people. For example, CAGs can be used to monitor earthquakes and protect residents’ safety.

In this paper, a home-made, miniaturized CAG is developed and integrated with a truck-borne system based on it for measuring absolute gravity in the field, and the gravity distribution characteristics of Xianlin reservoir in Hangzhou of China are also measured. This system’s internal coincidence accuracy and the external coincidence accuracy are evaluated. The internal coincidence accuracy, which is commonly known as precision, is expressed by the standard deviation of residuals between the gravity data of two repeated CAG measurements. The external coincidence accuracy, which is generally called accuracy, is expressed by the standard deviation in the residual of gravity between the gravity data measured by CAG and CG-5. Moreover, we have theoretically calculated the gravity values of the different measured points around the reservoir. The feasibility, reliability and practicality of the home-developed system are verified by comparing the theoretical results with the measured results. The home-made truck-borne CAG system is expected to improve the efficiency of the gravity being surveyed in the field.

## 2. Structure and Principle of the Truck-Borne System Based on CAG

The truck-borne system for measuring absolute gravity based on CAG is shown in Figure 1b. It consists of a miniaturized CAG and a truck-borne stabilization system. The CAG includes two parts: vacuum sensor system and control system (see Figure 1a). The diameter *D*, the height *H* and the weight *W* of this vacuum sensor system are 52 cm, 55 cm and 70 kg, respectively. The atom of rubidium 87 is used, and a loading rate of about 8 × 10^9^/s was realized in a compact quartz vacuum chamber. To better adapt to the measurement environment in the field, we optimized the vacuum sensor system, which is described in Ref. [37], such as a more compact and robust optical path, much smaller ion pump, more reliable support structure. The control system contains a laser system, an electronic control system. The scheme of the laser system is described in Ref. [38]. As the temperature and humidity vary widely in the field (when the air conditioner is turned off intermittently), a temperature control system was added, whose controlled temperature is 0.1 °C. Furthermore, the optical–mechanical structure was optimized and corresponding damping measures were taken in order to reduce the influence of vibration and shock so that the laser system could work stably in the field. The home-made electronic control system uses modular designs, which were adopted to reduce the size, weight and the power consumption of the control circuit. The size and weight of the control system were 18 U and 200 kg, respectively. The power consumption was about 250 W. The truck-borne stabilization system contains a three-axis leveling platform, which was used to level the attitude of the gravity sensor head, a passive vibration isolation platform, which was used to isolate high-frequency vibration noise from the vehicle, a differential GPS height measurement system, an uninterruptible power supply system (UPS) and a vehicle air conditioning temperature control system. Before carrying out the measurement, the truck was first parked at the measured point, the vibration isolation platform was suspended and the vacuum sensor system was adjusted to a horizontal position using the three-axis leveling platform. A large-range inclinometer was installed and used for rough leveling. In addition, a small-range, high-precision inclinometer was integrated in the vacuum sensor system, which can indicate the horizontal position. However, the inclinometer readings indicated the horizontal position should be calibrated in advance. Afterward, one-key adjustment of the tilt can be realized with the adjustment platform by using an automatic computer control program, and the accuracy was generally in the order of μrad. Then, the CAG was activated and began to measure the gravity data.

The basic working principle of CAG has been described in many articles [20,34,35,39]; here, we provide only a brief introduction. The free-fall object used in this experiment was a mass of ^87^Rb atoms cooled by lasers. The atoms were first prepared in a two-dimensional magneto-optical trap (MOT) and then transported to a three-dimensional MOT for loading, as shown in Figure 2a. Within 280 ms, the number of prepared atoms reached 10^8^. Through the process of polarization gradient cooling (PGC) further cooled their temperature to 5 μK. Then, they were selected at the quantum state of |*F* = 1, *m_F_* = 0>, an insensitive state to the magnetic fields in the first order. A sequence of three pulses of π/2–π–π/2 realizes the atom interferometer. Here, these three Raman pulses act as beamsplitters and mirrors, separating, redirecting and recombining the atomic wave-packets. The Raman π pulse duration was 10 μs and the time interval of *T* between Raman pulses was 55 ms. One measurement period took only 500 ms. Finally, the interference fringe in Figure 2b was obtained by scanning the chirp rate of α for Raman laser. With a state-selective fluorescence detection at the two atomic interferometer outputs, the transition probability P of atoms can be obtained by *P* = *P*_0_ − *C*/2 × cos(Δ*φ*), where *P_0_*, *C* and Δ*φ* represented the offset, contrast and phase of the atomic interference fringes, respectively. Deducing from the measurements of atomic state, we can obtain Δ*φ* = (*k*_eff_
*g* − *α*)*T*^2^, where *k*_eff_ is the effective wave vector of Raman beam, and *g* is the gravitational acceleration.

We evaluated the performance of this CAG in the laboratory by continuously measuring the gravity data over 7 days before carrying out truck-borne measurements. The gravity variations over time were observed. The results show that the measured gravity data obtained with the CAG and the theoretically calculated curve with the tidal model are highly consistent, and this CAG has a high stability. The measurement sensitivity of CAG is estimated to be about 300.0 $\mu \mathrm{Gal}/\sqrt{\mathrm{Hz}}\;$with the calculation of Allan deviation of measured gravity data, as shown in Figure 3. The actual measurement sensitivity is limited by several large noises, such as detection noise, vibration noise, laser phase noise, which were evaluated to be about 194.0$\;\mu \mathrm{Gal}/\sqrt{\mathrm{Hz}}$, 195.0$\;\mu \mathrm{Gal}/\sqrt{\mathrm{Hz}}$, 115.0$\;\mu \mathrm{Gal}/\sqrt{\mathrm{Hz}}$, respectively. Besides, the long-term stability of 4.0 μGal could be reached with an integration time of 10^4^ s. These parameters indicate the instrument’s potential for field applications.

## 3. The Results of Reservoir Gravity Measurement and the Analysis

The application of this home-made truck-borne CAG system to gravity surveys of the reservoir can verify the measurement performance of this system in the field. Here, we apply this to the Hangzhou Xianlin Reservoir (a newly built, large-scale reservoir); the measured route and points around the reservoir are shown in Figure 4. Points 4 and 5 are located on the reservoir dam, while the measured points 6, 7 and 8 are under the reservoir dam. The measured points 1, 2 and 3 are located on other places around the reservoir. The advantage of measuring gravity at almost the same or different heights is that the performance of our CAG can be fully tested, and the gravity distribution characteristics around the reservoir can be effectively obtained.

To evaluate the internal coincidence accuracy of this instrument in the field, we set the effective measurement time of each measured point to 20 min, and created a measured route from point 1 to point 8, along with an inverse route from point 8 to point 1 (see Figure 4). It is worth noting that the gravity values mentioned in this article are all relative values with a reference gravity value of −979318860.4 μGal. The round-trip measuring results from the CAG are shown in Figure 5; the black and red dots represent the results of two measurements, respectively. The results corrected the influence of the environment on gravity measurements, such as the Coriolis effect, the tides, the air pressure and the polar motion. The correction regarding the influence of the Coriolis effect was obtained by fitting a sine curve to the measured data based on the recorded heading angle of each measured point in the field. The maximum and minimum correction amounts of the eight different measured points can reach about 25.0 μGal and 5.0 μGal, respectively (See Figure 6). The change in gravity value caused by air pressure was corrected with the data collected by one high-precision barometer, and is around 3.0 μGal. The corrections to polar motion were calculated by inputting polar parameters, and were around 6.3 μGal.

As illustrated in Figure 5, the results of the two measurements were basically consistent and the residual reached up to 79.8 μGal. The internal coincidence accuracy of this instrument was evaluated to be 35.4 μGal by calculating the standard deviation of the residuals.

To further evaluate the external coincidence accuracy of this system, we compared the results with the gravity obtained by a relative gravimeter of CG-5 and the gravity reference point of high-precision absolute-gravity reference point, whose uncertainty was about 5.0 μGal in Zhejiang University of Technology (calibrated by a FG-5 absolute gravimeter). The measurement results are shown in Figure 7: the black dots represent the average of two CAG measurements and the red dots represent the results of the average of two CG-5 measurements. As shown in Figure 7, the results of the two measurements are basically consistent and the residuals reach up to 147.7 μGal. The system’s external coincidence accuracy was evaluated to be 76.7 μGal by calculating the standard deviation of the residuals.

In addition, it is obvious that the gravity values of the eight measured points are quite different due to their different geological structure. Considering that the latitude and longitude of the eight measured points are basically the same, the main reason for the difference in gravity should be the elevation of the measured points. Figure 8 shows the relationship between the elevation and the gravity of the measured points which obtained by CAG (the black dots) and CG-5 (the red dots). The average gravity gradient around the surface of reservoir was evaluated to be −215.8 μGal/m (CAG) and −215.1 μGal/m (CG-5). These results are consistent, but much smaller than the free-air gravity gradient. Meanwhile, the gravity of points at a higher elevation was not linearly correlated with the elevation. The reasonable explanation for this is that the geological structure around the reservoir is complex and the GPS accuracy when measuring altitude in our experiments is about 0.5 m. To further verify this guess and the reliability of the truck-borne CAG system, we theoretically calculated the gravity values of eight measured points.

The gravity of the eight measured points was calculated using the following formula [40]:(1)$$g_{cal}{=g}_{n\;}{+g}_{fa}+g_{sb}+g_{tc},$$
which considers all known factors that contribute to the difference between gravity observation at different locations. The normal gravity $g_{n}$ is calculated with the Somigliana equation [41]:(2)$$g_{n}{=g}_{e}\left(\frac{{1+k\sin}^{2}\lambda }{\sqrt{{1\;-\;e}^{2}\sin^{2}\lambda }}\right),$$
which determines the normal gravity value of a point on the surface of the Geodetic Reference System 1980 (GRS80) ellipsoid, with $g_{e}=9.7803267715$ m/s^2^ is the normal gravity at equator, k=0.001931851353 is the normal gravity constant, $e^{2}=0.00669438002290$ is the square of the first numerical eccentricity, and λ is the geodetic latitude of the observation point. The free-air correction $g_{fa}$ accounts for gravity observations that were not made on the surface of the ellipsoid, which is essentially a correction to the observed gravity for the inverse-distance-squared decay of gravity on moving away from the Earth [42]. Applying a linear approximation based upon a spherical Earth model, the free-air correction can be computed as (unit in mGal, 1 mGal = 10^−5^ m/s^2^):(3)$$g_{fa}=0.3086H,$$
with H the height of the observation above the ellipsoid in meters.

The remaining two terms, $g_{sb}{+g}_{tc},$ is called the complete Bouguer correction [1], which accounts for the additional mass between the observation level and the referenced datum. The simple Bouguer correction term, $g_{sb},$ approximates all mass above the datum with a homogeneous, infinitely extended slab of thickness H and a typical crustal density of 2670 kg/m^3^ and can be computed as (unit in mGal):(4)$$g_{sb}=0.1119H,$$

The terrain correction term $g_{tc}$ accounts for the topography’s departures from the simple plate approximation made by the simple Bouguer correction, which is essential in rugged areas, and can be evaluated using a prism-summation algorithm [43] based on the 1 m × 1 m Digital Elevation Model (DEM) covering the reservoir area. The fortran code for computing the gravitational attraction of a single rectangular prism is provided in appendix B of Blakely’s 1996 book [40]. The calculated results are shown in Figure 9a. Their variation trends are consistent, although there is a difference of about 4.5 × 10^3^ μGal between the measured and the calculated gravity values. This differences may result from the gravity anomaly caused by the deep-large-scale low density structure around the reservoir, the flaws of the computational models and the inaccuracies of the geological parameters. Moreover, both the measured and calculated gravity values of 8 measured points are relative to the reference gravity value of point 2, and they are almost the same, which could be seen from Figure 9b. Besides, the deviation between measured value and the calculated value did not exceed 600 μGal. The main reason for the differences between the theoretical and experimental results are that the geometry model of the local topography is imperfect when assuming the density of the whole topographic mass *ρ* = 2670 kg/m^3^ and the GPS accuracy for measuring altitude in our experiments was approcimately 0.5 m.

## 4. Conclusions

In this paper, we set up a truck-borne system based on a miniaturized CAG to measure absolute gravity in the field. To verify the system measurements, we applied the system to the gravity survey of Xianlin reservoir. The measured results show that this system has some excellent characteristics, including high reliability, strong environmental adaptability and high stability. According to the measured results, the internal and external coincidence accuracy was evaluated to be 35.4 μGal and 76.7 μGal, respectively. In addition, the gravity values of the measured points were theoretically calculated. The results of relative gravity values are in good agreement with the measured results, although there is a difference of about 4.5 × 10^3^ μGal. These differences may result from the gravity anomaly caused by the deep–large-scale, low-density structure around the reservoir, the flaws of the computational models and the inaccuracies of the geological parameters. Therefore, this truck-borne system based on a miniaturized CAG can improve the efficiency of gravity mapping in the field.

However, the performance of this truck-borne CAG system can be further improved. The main factor that affects the accuracy of the system is the measurement sensitivity, which can be improved by increasing the length of the interferometry zone and optimizing the laser phase noise.

After improving the performance, it is expected that the system could provide support to truck-borne gravity-mapping, geological body interpretation and geophysical research. It may be feasible to monitor the gravity anomalies that occur around the reservoir in the future.

## Figures and Tables

**Figure 1 sensors-22-06172-f001:**
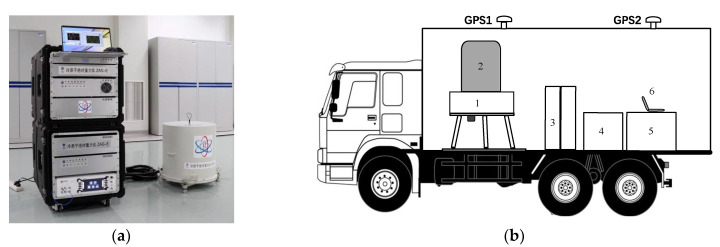
The truck-borne system based on CAG for measuring absolute gravity in the field: (**a**) The apparatus of CAG; (**b**) The truck-borne system includes: 1. three-axis leveling platform and the passive vibration isolation platform; 2. the vacuum sensor system; 3. UPS; 4. optical system; 5. control system; 6. computer.

**Figure 2 sensors-22-06172-f002:**
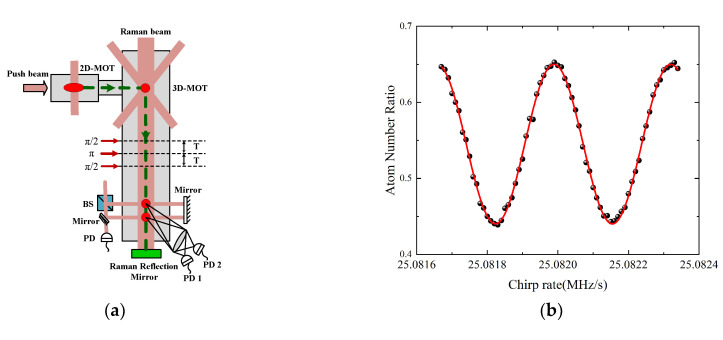
The measuring principle of CAG atomic gravimeter. (**a**) The working principle of gravity sensor head; (**b**) the cold atom interference fringes obtained in the laboratory.

**Figure 3 sensors-22-06172-f003:**
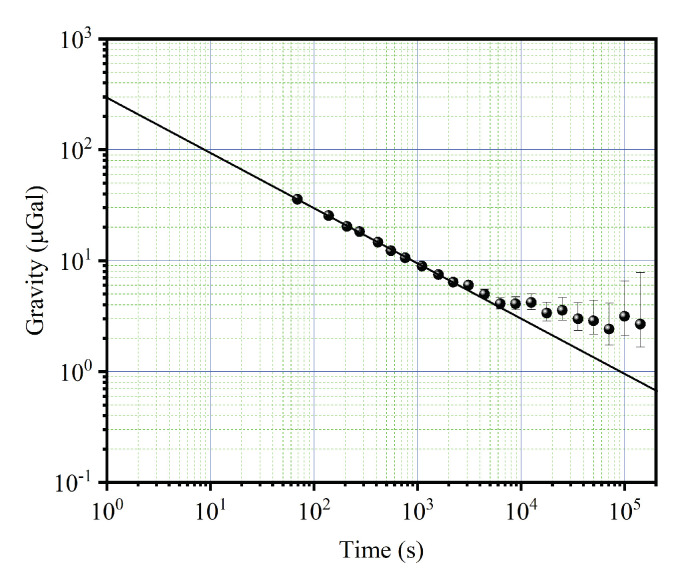
The Allan deviation of the measured gravity data.

**Figure 4 sensors-22-06172-f004:**
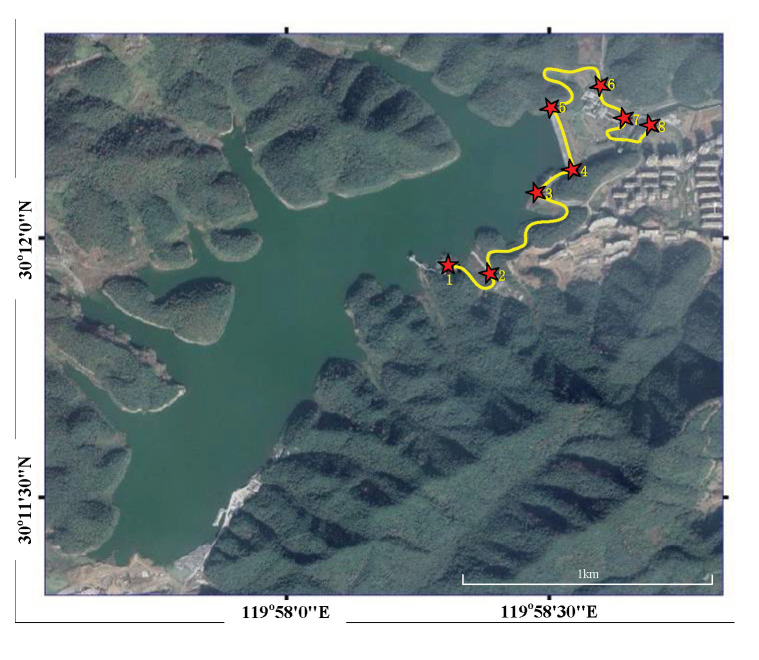
The route and 8 points of gravity measurement around Xianlin reservoir.

**Figure 5 sensors-22-06172-f005:**
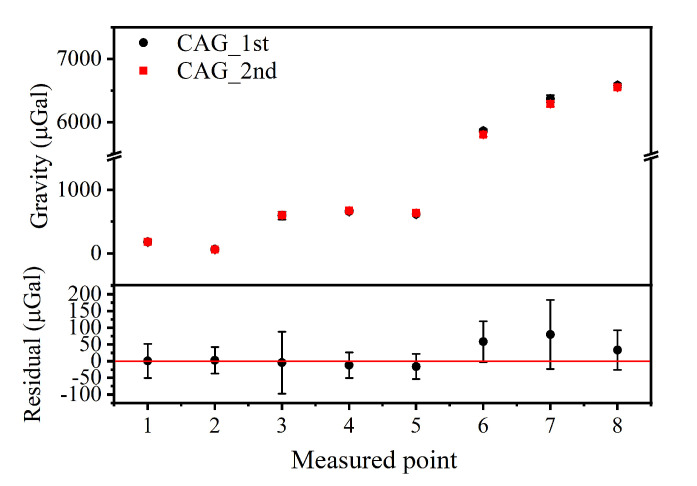
The gravity data of 8 measured points for two measurements (the black and red dots represent the results of first and second measurements, respectively) with the home-made, truck-borne CAG system, the residual data are also shown in the figure below.

**Figure 6 sensors-22-06172-f006:**
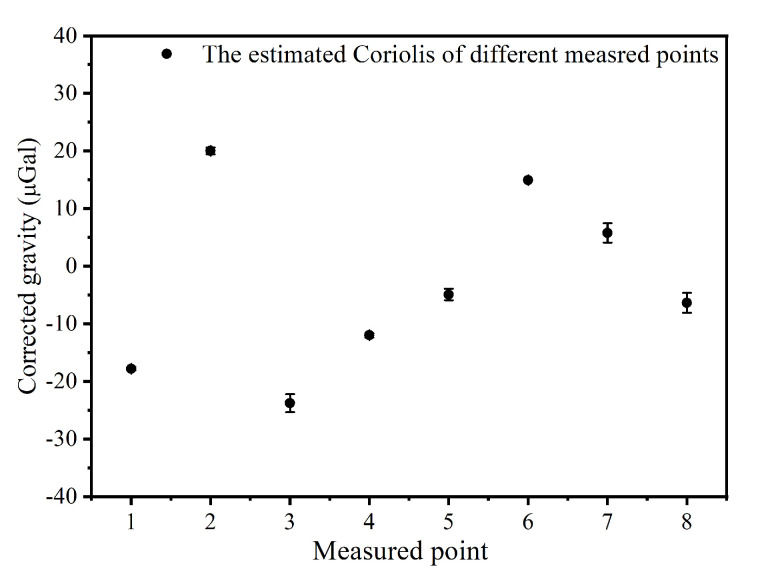
The corrected gravity of different measured points due to Coriolis effect.

**Figure 7 sensors-22-06172-f007:**
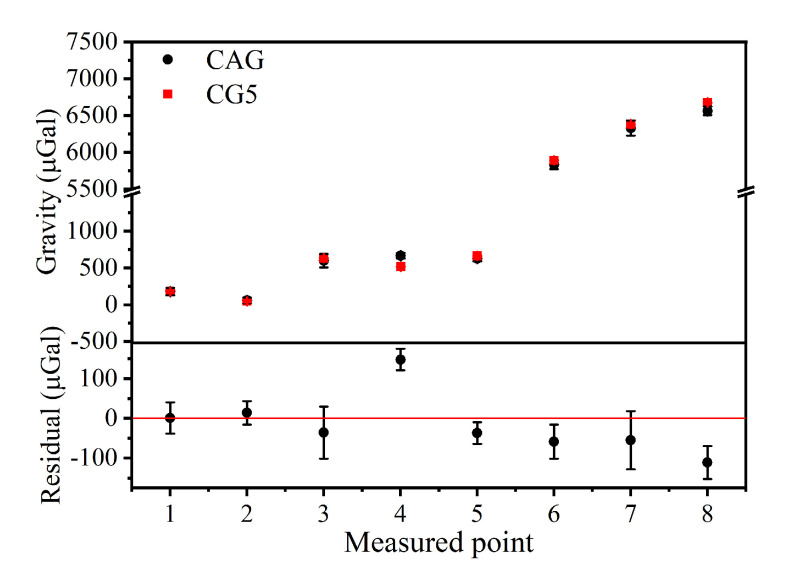
The gravity data of 8 measured points with home-made, truck-borne CAG system and the relative gravimeter of CG-5 (the black and red dots represent the results of CAG and CG-5, respectively); the residuals data between them are also shown in the figure below.

**Figure 8 sensors-22-06172-f008:**
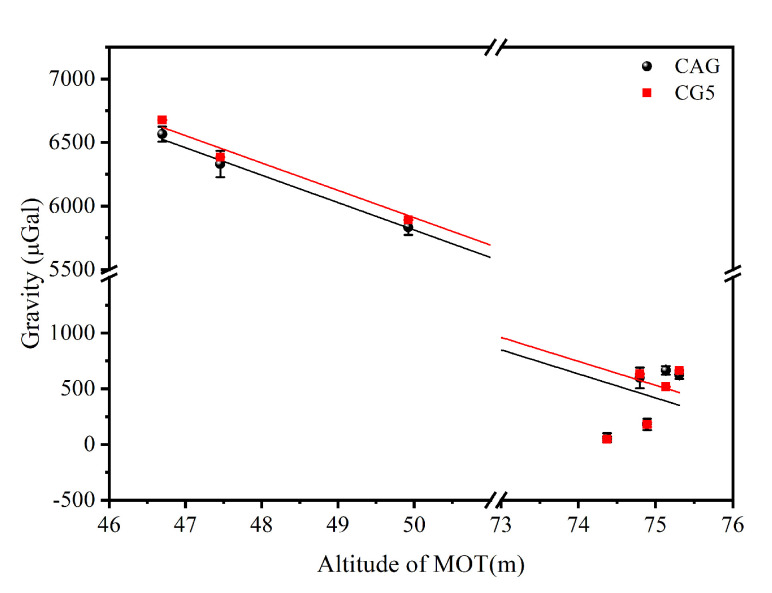
The relationship between the elevation and the gravity of measured point which obtained by CAG (the black dots) and CG-5 (the red dots) respectively.

**Figure 9 sensors-22-06172-f009:**
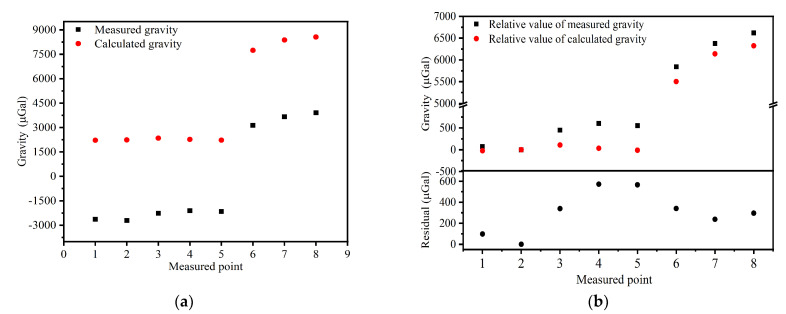
(**a**) Comparison between the gravity measurement results with CAG (the black dots) and the calculation results (the red dots) around the Xianlin reservoir, (**b**) comparison between the gravity measurement results with CAG (the black dots) and the calculation results (the red dots) around the Xianlin reservoir with a reference of the gravity value of point 2.

## Data Availability

Not applicable.

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
