# Peer review of "A Truck-Borne System Based on Cold Atom Gravimeter for Measuring the Absolute Gravity in the Field"

_sensors, 2022, doi:10.3390/s22166172_

Round 1

Reviewer 1 Report

The article by Wang et al. present a mobile, home-made CAG, and use it to characterize the gravity near Xianlin reservoir. The work is interesting, and represents a major step towards realistic applications of CAG. However, before I can recommend publication, I would like to raise a couple of questions to the authors:
(1) Could the author provide more details on the measurement results. This includes characterization of the sensitivity [Gal/sqrt(Hz)], and long-time stability at a fixed spatial point. 
(2) The measured gravity deviates from a linear scalling with the altitude of the eight points. And the author provides a possible explanation that the geological structure around the reservoir is complex. To this end, is it possible for the authors to perform some extra measurement at several spatial points at a place with simple geological structure, and observe the expected linear scaling of gravity with elevation? That results will represents a more convincing calibration of the reported CAG.

Author Response

     Thank you for reviewing our manuscript. We have revised the manuscript carefully according to your valuable suggestions. Point by point responses to the your constructive  comments are listed .Please see the attachment.

Reviewer 2 Report

Dear Authors,

Thank you for your efforts to improve measuring gravity field. The manuscript is well written, theoretically is correct and practically is tested on real measured data. The repeated measurements satisfy and confirms the stability of the proposed system. The power consumption is 76 about 250 W which is accepted in the field work. Moreover, the authors tested the change of gravity value caused by air pressure. The correction on polar motion is also calculated.

regards

Author Response

    Thank you very much for the comments of the reviewer. In order to improve the quality of our manuscript furtherly, we have revised our paper carefully and thoroughly, and some additional modifications have been made. Please see the attachment.

Reviewer 3 Report

My apologies for the delayed review. I hope the level of detail will help offset the delay. Please see attached report. 

Author Response

(The authors gave the same response as above.)

Round 2

Reviewer 1 Report

I think the manuscript has been sufficiently improved, and warrant publication.

Author Response

Thank you very much for the comments of the reviewer. In order to improve our manuscript furtherly, we have checked the paper again, and made some new modifications, please see the attachment.

Reviewer 3 Report

Please see attached report. Apologies for the delay. 

Author Response

According to the suggestions of the reviewer, we have checked the paper again, and made some new modifications, please see the attachment.
